# Infrared beam-shaping on demand via tailored geometric phase metasurfaces employing the plasmonic phase-change material In$_3$SbTe$_2$

Lukas Conrads [1,5] ✉, Florian Bontke[1,5], Andreas Mathwieser [2], Paul Buske [3], Matthias Wuttig [1], Robert Schmitt[2], Carlo Holly [3,4] & Thomas Taubner [1] ✉

Conventional optical elements are bulky and limited to specific functionalities, contradicting the increasing demand of miniaturization and multi-functionalities. Optical metasurfaces enable tailoring light-matter interaction at will, especially important for the infrared spectral range which lacks commercially available beam-shaping elements. While the fabrication of those metasurfaces usually requires cumbersome techniques, direct laser writing promises a simple and convenient alternative. Here, we exploit the non-volatile laser-induced insulator-to-metal transition of the plasmonic phase-change material In$_3$SbTe$_2$ (IST) for optical programming of large-area metasurfaces for infrared beam-shaping. We tailor the geometric phase of metasurfaces with rotated crystalline IST rod antennas to achieve beam steering, lensing, and beams carrying orbital angular momenta. Finally, we investigate multi-functional and cascaded metasurfaces exploiting enlarged holography, and design a single metasurface creating two different holograms along the optical axis. Our approach facilitates fabrication of large-area metasurfaces within hours, enabling rapid-prototyping of customized infrared meta-optics for sensing, imaging and quantum information.

Metasurfaces composed of metallic or dielectric nanoantennas of subwavelength size have been widely used to tailor the properties of scattered light, such as amplitude, phase, and polarization[1,2]. Numerous antenna geometries with varying sizes and shapes exploiting multiple resonance modes have enabled various functionalities such as beam steering[3], lensing[4], or holography[5]. However, full 2π phase control often requires sophisticated antenna designs with complex shapes and multiple resonance modes. Another promising approach to alter the phase of the scattered light relies on employing geometrically rotated antennas of the same size in combination with circularly polarized light[6]. Accordingly, geometric phase, also called Pancharatnam–Berry phase, metasurfaces facilitate applications such as non-linear optics[7], manipulation of orbital angular momentum[8], and spin-selective deflection[9,10].

The fabrication of these metasurfaces usually relies on cumbersome and complex fabrication techniques involving multiple lithography and etching steps. Direct laser writing of metasurfaces hence offers great potential for speeding up the fabrication and enabling rapid prototyping of various metasurfaces with different functionalities[11].

[1]Institute of Physics (IA), RWTH Aachen University, D-52056 Aachen, Germany. [2]Fraunhofer Institute for Production Technology IPT, 52056 Aachen, Germany. [3]Chair for Technology of Optical Systems, RWTH Aachen University, 52056 Aachen, Germany. [4]Fraunhofer Institute for Laser Technology ILT, 52056 Aachen, Germany. [5]These authors contributed equally: Lukas Conrads, Florian Bontke. ✉e-mail: conrads@physik.rwth-aachen.de; taubner@physik.rwth-aachen.de

Among others[12], promising materials for direct laser writing of metasurfaces are phase-change materials (PCMs)[13]. These materials exhibit at least two (meta-) stable phases, the amorphous one and the crystalline one, which differ significantly in their optical and electrical properties[14,15]. The strong refractive index contrast between both phases is attributed to another bonding mechanism in the crystalline phase, called metavalent[16–19]. This makes PCMs prime candidates for non-volatile dynamic metasurface tuning based on a change in the refractive index[20,21]. In the past years, numerous applications such as active beam steering[9,22], lensing[9,13], and shaping thermal emission[23] have been demonstrated.

Recently, the PCM In$_3$SbTe$_2$ (IST) has been introduced, which shows a Drude-like behavior with a negative real part of the permittivity in its crystalline phase for the entire infrared spectral range[24]. Therefore, it is classified as a plasmonic PCM enabling direct writing of metallic nanostructures by locally crystallizing the PCM with precise laser pulses. Active resonance tuning can be achieved by modifying the antenna structures themselves and allows for reprogramming once written metasurfaces. Not only tuning antenna resonances by reconfiguring the antenna geometries[25–28], but also active metasurfaces for thermal emissivity shaping[29–31] and tailoring polariton responses[32–35] have been demonstrated.

However, rapid-prototyping of complex IST metasurfaces for real-world applications has not been shown yet. Tailored phase-modulated metasurfaces with the plasmonic PCM IST promise arbitrary beam-shaping in the infrared spectral range, which inherently lacks commercially available devices.

Here, we take advantage of the interplay of circularly polarized light with rotated rod antennas to encode multiple metasurfaces optically written directly into IST. We use a commercial direct laser-writing system for programming large-area metasurfaces and investigate their respective functionalities. First, we demonstrate beam steering metasurfaces with varied supercell periods to obtain different beam deflection angles. Second, we design and investigate a metalens with a focal length of 11.5 cm. Third, we exploit the orbital angular momentum of light and verify the mediated topological charges by revealing the spiral intensity pattern. Then, the phase profile of a metasurface hologram is designed with the Gerchberg–Saxton algorithm and combined with the phase profile of a magnifying lens, highlighting an easy way for combining multiple functionalities within a single metasurface. Cascading two different metasurfaces allows the combination of different functionalities by exploiting the unaffected incident light. Finally, we use a diffractive neural network for designing a single hologram metasurface featuring two different hologram patterns at certain distances behind the metasurface.

## Results

### Programming large-area geometric phase beam steering metasurfaces

The conventional fabrication of large-area metasurfaces is complex and requires several lithography steps, incompatible with the requirements of rapid prototyping. Direct laser writing of metallic antennas with the plasmonic PCM IST instead offers a promising platform for the rapid development of complex phase-modulated metasurfaces. The large-area metasurfaces investigated are directly optically written in 100 nm amorphous IST on top of a transparent CaF$_2$ substrate with precise laser pulses (see Fig. 1A). In particular, we applied the direct laser writing system Photonic Professional GT from Nanoscribe equipped with highly precise galvo mirrors to redirect the laser beam on the sample and induce the crystallization process (see Methods). While amorphous IST exhibits dielectric behavior with a constant permittivity of 14, its crystalline phase follows a metallic Drude-like behavior ($\varepsilon' < 0$) in the entire infrared spectral range[24]. The real part of the permittivity for amorphous and crystalline IST is shown in Fig. 1B (see Supplementary Note 1 for more details). A comparison of

the permittivity of the plasmonic PCM IST with dielectric PCMs such as Ge$_3$Sb$_2$Te$_6$ and Ge$_2$Sb$_2$Se$_4$Te$_1$ and the phase transition material VO$_2$ is shown in Supplementary Note 2. Consequently, it is possible to directly program entire large-area metasurfaces by locally crystallizing spatially varying nanoantennas within the amorphous IST. Full 2π-phase control within the metasurfaces fabricated is achieved by rotating the antennas from zero to 180 degrees and illuminating the metasurface with circularly polarized light. This concept is also known as the Pancharatnam–Berry phase or geometric phase. The rotation angle $\beta$ directly translates to the phase of the scattered light $\phi$ via:

$$\phi = 2 \cdot \beta \qquad (1)$$

The phase is only controlled by the rotation of the antenna and the chirality of the scattered light is reversed with respect to the incident light, allowing for a clear distinction of the incident light from the scattered light[36]. A more detailed description of this geometric phase concept can be found in Supplementary Note 3.

The employed antennas featuring a length of 2.5 μm are designed to be resonant at a wavelength of 9 μm, which is the operation wavelength of the infrared quantum cascade laser and the employed quarter-wave plates. Utilizing the resonance wavelength of the nanoantennas ensures maximum scattering and, consequently, maximum efficiency of the metasurfaces investigated. Note, that this choice of wavelength does not display a limit of the demonstrated concept and any other infrared wavelength would be also possible. Measured transmittance spectra of crystallized IST antennas for different lengths can be found in Supplementary Note 4.

First, we investigate two beam steering metasurfaces with spatially varying antennas along the supercell period Γ. Within the supercell period, the phase gradient varying from 0 to 2π is determined by the rotation angle of the antennas. Engineering the supercell period Γ leads to the deflection angle via[3]:

$$\theta = \arcsin \frac{\lambda}{\Gamma} \qquad (2)$$

The operation wavelength is set to 9 μm and two metasurfaces with supercell periods of 18 and 36 μm are designed, leading to theoretically calculated deflection angles of 30° and 14.5°, respectively. The period between individual antennas is set to 4 μm with an antenna length and width of 2.5 and 0.7 μm, respectively. The height of the antennas is limited by the thickness of the IST layer, resulting in a maximum antenna height of 100 nm. Light microscope images of the optically crystallized metasurfaces are shown in Fig. 1C. Note that for the beam steerer with a corresponding supercell period of 18 μm, two adjacent supercells are displayed. The entire metasurface with a size of 4 × 4 mm² consists of 1 million individual antennas with varied orientations fabricated within 2.5 h. The phase difference introduced between adjacent antennas is given by 2π/N, with N referring to the number of antennas within the supercell.

Afterward, we characterize the deflected beam transmitted through the metasurfaces with a home-build setup (see Methods). The incident left-handed circularly polarized (LCP) beam is transmitted through the metasurface, while the scattered right-handed circularly polarized (RCP) beam is deflected according to formula 2. The detector is rotated to measure the angle-resolved beam intensity. Applying a second quarter-wave plate combined with a linear polarizer allows for clear distinction between both polarization chiralities.

Figure 1D displays the measured laser intensities after passing both beam steering metasurfaces dependent on the angle and polarization chirality. The initial LCP chirality (dashed curves) is transmitted through the metasurface, showing maximum intensity at 0° deflection. In contrast, light with the opposite RCP chirality (solid lines) is deflected according to formula (2). Here, the metasurface with a

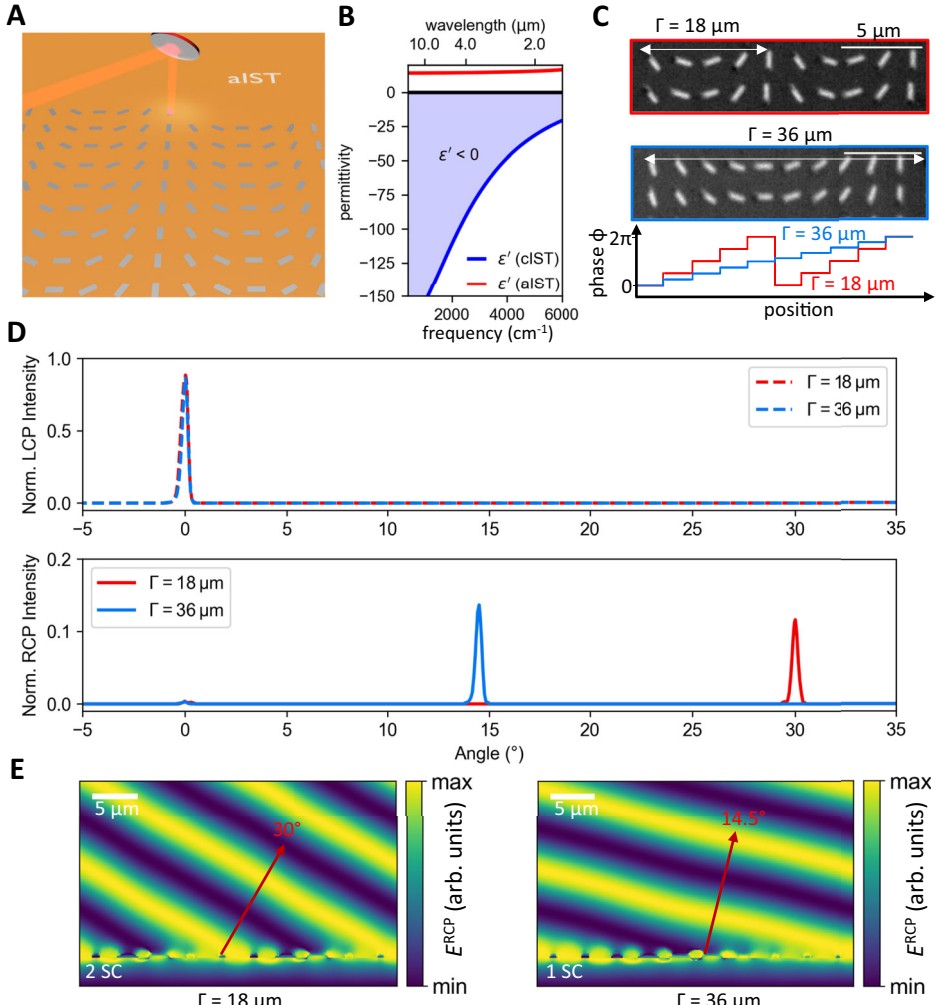

**Fig. 1 | Beam steering metasurface with IST. A** Concept of programming large-area geometric metasurfaces by optically crystallizing rotated IST rod antennas. Controlling the rotational angle of the rod antennas results in full 2π phase control of the metasurface. **B** Real part of the permittivity for amorphous and crystalline IST. In the crystalline phase, IST shows metallic behavior (ε' < 0). **C** Light microscope images of two different beam steering metasurfaces with supercell periods Γ of 18 μm and 36 μm. The phase increases from 0 to 2π along the supercell period. The scale bars are 5 μm. **D** Measured beam intensities for two different beam steering metasurfaces with supercell periods of 18 and 36 μm. The RCP component of the circularly polarized light is deflected, while the initial LCP light is not affected. **E** Simulated RCP electric field for incident LCP light for the two metasurfaces. The measured beam deflection is well reproduced for both supercell (SC) periods.

supercell period of 18 μm exhibits a peak for the deflected RCP light at 30°, while the metasurface with the larger supercell period of 36 μm reveals a deflection angle of 14.5°. The experimentally measured beam intensities are in very good agreement with numerical far-field simulations (see Supplementary Note 5). The efficiency of the metasurfaces is determined by comparing the intensity of the deflected light with the unaffected transmitted light, revealing values around 10%. We attribute the higher efficiency of the metasurface with a larger supercell period of 36 μm to smaller phase increment steps due to the doubled amount of rotated antennas compared to the metasurface with a supercell period of 18 μm (c.f. Figure 1C bottom).

We performed field simulations of the RCP electric field transmitted through the metasurface for incident LCP light, as shown in Fig. 1E. The RCP component is deflected by 30° for the metasurface with a supercell period of 18 μm, and by 14.5° for the metasurface with a supercell period of 36 μm, respectively. Hence, the experimentally obtained deflection angles are well reproduced with electric field simulations and numerical far-field simulations.

Moreover, our designed beam steering metasurface is robust against fabrication imperfections due to the broad electric dipole resonances of the IST antennas. We demonstrate that even length

variations of ±20% perform similarly without a significant decrease in performance (see Supplementary Note 6).

### Focusing infrared radiation with a metalens

In addition, we investigate a metalens consisting of rotated crystalline IST rod antennas which focus the converted RCP radiation to a focal spot 11.5 cm behind the metasurface (see sketch in Fig. 2A). The large focal distance of the metalens is chosen arbitrarily to ease subsequent measurement by ensuring enough distance between the metasurface and the focal spot. The applied phase pattern of the metasurface is shown in Fig. 2B, featuring concentric rings of equal phases calculated with:

$$\phi(r) = \frac{2\pi}{\lambda}\left(\sqrt{r^2 + f^2} - f\right) \tag{3}$$

Here, $f$ denotes the focal length of 11.5 cm and the operation wavelength $\lambda$ of the metasurface is again set to 9 μm, while $r$ determines the radial antenna position.

Figure 2C displays a photograph of the fabricated 8 × 8 mm² large-area metasurface with each antenna resembling a fixed phase value.

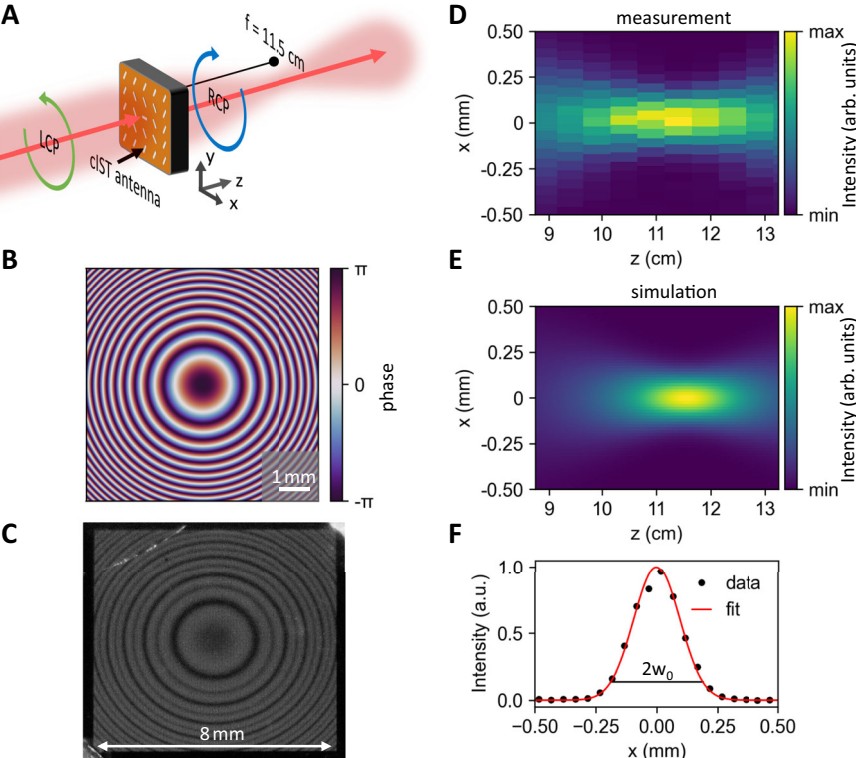

**Fig. 2 | Metalens with IST. A** Schematic sketch of the working principle applied. The incident LCP light is converted to RCP radiation and focused according to the designed focal length. **B** Imprinted phase profile of the metalens consisting of concentric rings with equal phases. **C** Photograph of the fabricated $8 \times 8\,mm^2$ metasurface consisting of 4 million rotated IST antennas. **D** Measured intensity distribution in the xz-cross section clearly displaying a focal spot 11.5 cm from the metasurface. **E** Simulated intensity distribution of a metasurface with the applied phase pattern from (**B**). **F** Cross-section of the measured focal spot revealing a beam waist $w_0$ of 189 μm.

The period between adjacent antennas is set to 4 μm, and the exact orientation of each antenna at different positions on the metasurface is calculated with Eq. 3, leading to a nearly continuously varying phase pattern. The entire metasurface consists of 4 million single antennas fabricated in 8 h. The same procedure is used for all the following metasurfaces.

The corresponding intensity distribution measured in the xz-cross-section is shown in Fig. 2D (see Methods about the measurement procedure). A clear focal spot with maximum intensity is observed at 11.5 cm from the metasurface. Our experimental results are supported by numerical simulations done with the Python package LightPipes by simulating the propagation of the incident Gaussian beam transmitted through the metasurface with the phase pattern of Fig. 2B. The simulations shown in Fig. 2E validate the focal spot at 11.5 cm from the metasurface with a comparable beam waist diameter in the focal spot. Finally, we investigate the diameter of the focal spot in Fig. 2F exhibiting a beam waist of 189 μm by fitting a Gaussian function to the measured intensity. The large beam waist is caused by the intrinsic ultra-low numerical aperture (NA) of 0.03 due to the large focal length of the metasurface. The low NA is only chosen for simplified measurement of the focal spot. Our approach of patterning IST metasurfaces allows for metalenses with larger NA too. The retrieved beam parameter product of the laser after passing the metasurface is close to the optimal value, confirming that our metasurface preserves the intrinsic beam quality (see Supplementary Note 7).

## Beam-shaping exploiting orbital angular momenta

Generally, the concept of orbital angular momentum (OAM) has gained much interest in the past years[37]. The characteristic vortex beams consisting of different OAM modes feature a ring-like intensity distribution combined with helical phase factors $exp(il\phi)$, with $l$ referring to the topological charge and $\phi$ the azimuthal angle. The orthogonal OAM modes can be superimposed to increase information capacity and boost optical communication systems[38,39].

Here, we design three beam-shaping metasurfaces carrying an OAM in addition to the intrinsic spin angular momentum mediated by the polarization of the light. Therefore, we employ three helical phase patterns with topological charge $l$ of one, three and five (see Fig. 3A). As stated before, the orientation of the antennas is calculated according to the exact position onto the metasurface.

The corresponding far-field intensity measurements of our metasurfaces can be seen in Fig. 3B. The diameter of the observed rings increases, pointing towards different topological charges. The measured intensity cross-section along the rings can be found in Supplementary Note 8. The determination of the intrinsic OAM is achieved by direct interference of the unaltered beam with the light carrying the OAM after passing the metasurface (see Methods for more details). The results are shown in Fig. 3C. Here, the number of observable spiral arms is directly associated with the topological charge. The first image for l = 1 features only one spiral arm, while the second image for l = 3 features three spiral arms and so on, verifying the OAM carried by the photons after passing the metasurface.

Numerical simulations of a Gaussian beam imprinted with the corresponding phase profiles in Fig. 3A are shown in Fig. 3D. Characteristic ring-like patterns associated with doughnut modes with increasing diameter for increasing topological charges appear with diameters comparable to the experimentally obtained images. Figure 3E displays far-field intensity simulations of the OAM beam superimposed by the incident Gaussian beam, revealing the spiral

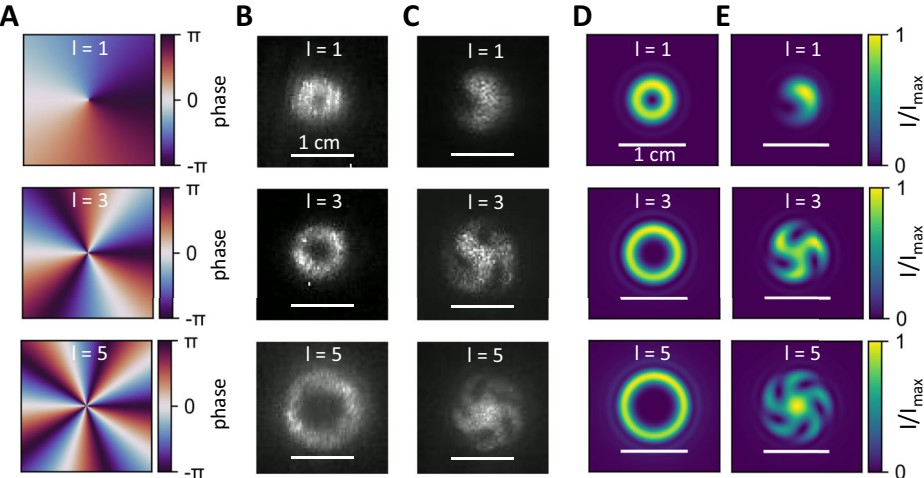

**Fig. 3 | Encoded orbital angular momentum. A** Applied helical phase patterns carrying an orbital angular momentum of 1, 3, and 5 for three different metasurfaces. **B** Measured far-field intensity patterns of the beam 60 cm from the metasurface displaying a ring-like structure with increasing diameter for higher orbital angular momenta. **C** Direct interference of the unaffected incident light with the scattered light from the metasurface leads to a spiral pattern with the number arms as described before. The simulations are in very good agreement with the experimental data.

**D** Simulated far-field intensity pattern of the helical phase profiles demonstrating similar ring patterns compared to (**B**). **E** Simulated spiral patterns due to the interference of the incident Gaussian beam and the transformed beam after passing the metasurface reveal very good agreement between experiment and simulation.

### Enlarged infrared holography with a multifunctional metasurface

The vast flexibility of direct programming complex phase patterns with IST is demonstrated by designing a holographic phase pattern with the Gerchberg–Saxton algorithm (see Methods) to create a specific infrared hologram in the far field. Because the size of the designed holographic image is small and, therefore, challenging to measure, we superimpose the retrieved phase pattern of the hologram with the phase pattern of a magnifying lens. This highlights the ability to combine multiple functionalities within a single metasurface and leads to simplified imaging of the resulting hologram with a size of several centimeters. The concept is sketched in Fig. 4A. The targeted far-field intensity pattern is displayed in Fig. 4B, showing the letters "ir nano" with uniform intensity. The final calculated phase pattern combined with the phase pattern of the lens for the designed hologram 60 cm from the metasurface can be seen in Fig. 4C. Here, the phase values are simply added to each other. The photograph of the fabricated 8×8 mm² metasurface shows a similar pattern caused by scattering of the visible light (c.f. Fig. 4D). The measured hologram in combination with numerical simulation is shown in Fig. 4E for different distances behind the metasurface. While for small distances (e.g., 20 cm behind the metasurface), no clear image is observed, at larger distances around 60 cm behind the metasurface, the letters "ir nano" are clearly resolvable. Due to the magnifying lens, the size of the hologram increases with further distance to the metasurface. The experimental data and the simulations agree well with each other.

In another experiment, a third functionality of a beam steerer is added to the magnified holography metasurface by superimposing the previous phase mask with the phase mask of a beam steerer (see Fig. 5A). The resulting metasurface deflects the enlarged hologram by a given angle of 10° as shown in Fig. 5B. For better visualization, the incident LCP light is not filtered out completely, leading to a point-like intensity pattern at 0°. This demonstrates the ability of combining several different functionalities within the same metasurface by simply superimposing the corresponding phase masks which is not possible with conventional optical elements.

Moreover, by cascading two metasurfaces the remaining unaffected incident polarized light can be reused for another functionality. This is possible because the converted RCP light from the first metasurface transmits unaffected through the second metasurface. A sketch of the setup and concept employed is shown in Fig. 5C by inserting another metasurface (black dotted line) into the beam path. The first metasurface employed corresponds to the OAM metasurface with l = 3, while the second metasurface features the deflected and enlarged hologram from Fig. 5A. The resulting intensity profiles at the screen are displayed in Fig. 5D. Here, the characteristic ring-pattern of the OAM appears at 0° and the deflected hologram at 10°. A second example of cascaded metasurfaces with two holograms is shown in Supplementary Note 9.

Finally, we design a hologram metasurface exhibiting two different holograms at different positions $z_1$ and $z_2$ from the metasurface with a diffractive neural network (see Supplementary Note 10)[40,41]. The phase of the scattered light is altered to not only display a hologram at a set distance, but also additionally enables the reordering of the light upon propagation to form a second hologram with a different intensity distribution. This concept is visualized in Fig. 6A. Here, at position $z_1$, the hologram showing a distorted lattice representing the amorphous phase with the caption "aIST" is displayed. At a second position $z_2$, the observable hologram changes to a periodic lattice representing the crystalline phase with the caption 'cIST'. The calculated phase mask is shown in Fig. 6B. Notice that we took the actual beam shape of the laser into account to achieve homogeneous intensity distributions within the hologram images. Measurements performed with a Pyrocam IV by *Ophir Photonics* at 16 and 21 cm are shown in Fig. 6C. The observable pattern changes as designed upon increasing the distance. Numerical simulations (see Fig. 6D) are in good agreement with the experimental data. The evolution of the hologram by varying the distance behind the metasurface is shown in Supplementary Video 1.

### Discussion

In summary, we demonstrated direct programming of geometric phase metasurfaces consisting of rotated crystalline IST rod antennas within the plasmonic PCM IST for infrared beam-shaping. Tailoring the phase of the metasurface gives access to numerous

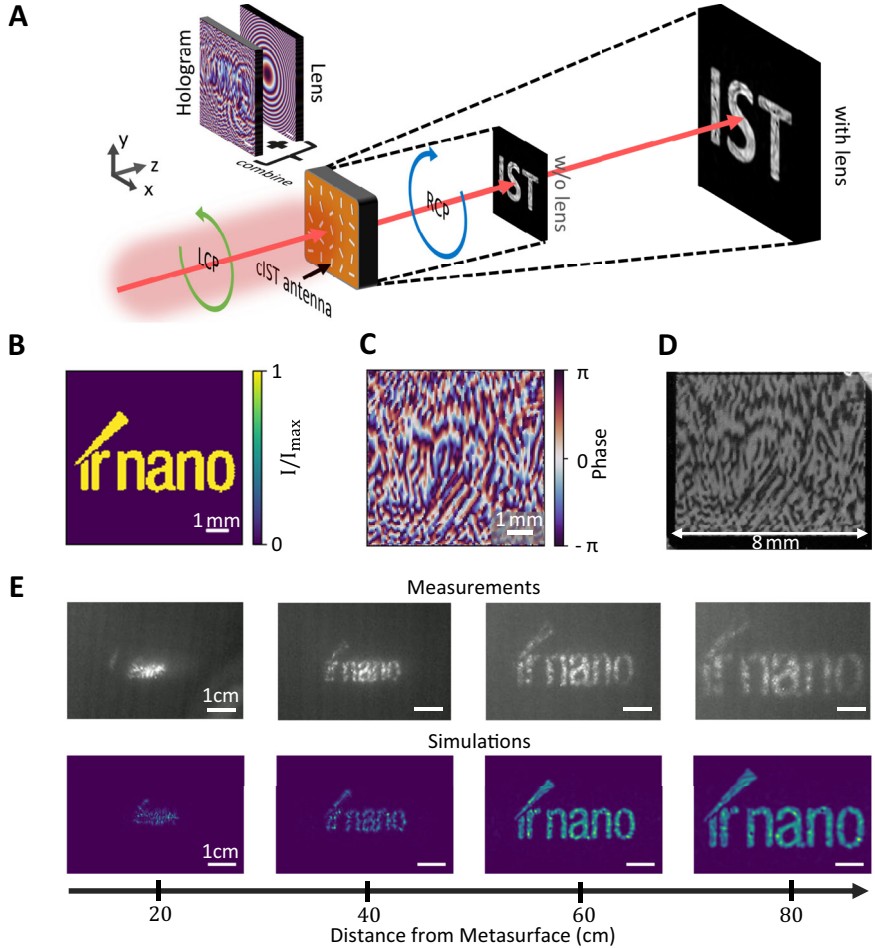

**Fig. 4 | IST holography metasurface. A** Schematic sketch of combining the phase pattern of the hologram with the pattern of a magnifying lens to achieve enlarged imaging of the RCP holographic image. **B** Targeted far-field intensity pattern displaying the letters "ir nano" at 60 cm distance from the metasurface. **C** Calculated metasurface phase profile with the Gerchberg–Saxton algorithm according to the targeted intensity pattern superimposed with the phase profile of a magnifying lens. **D** Photograph of the 8 × 8 mm² metasurface written optically. **E** Measured and simulated intensity patterns at varied distances from the metasurface. The designed hologram is best visible 60 cm behind the metasurface.

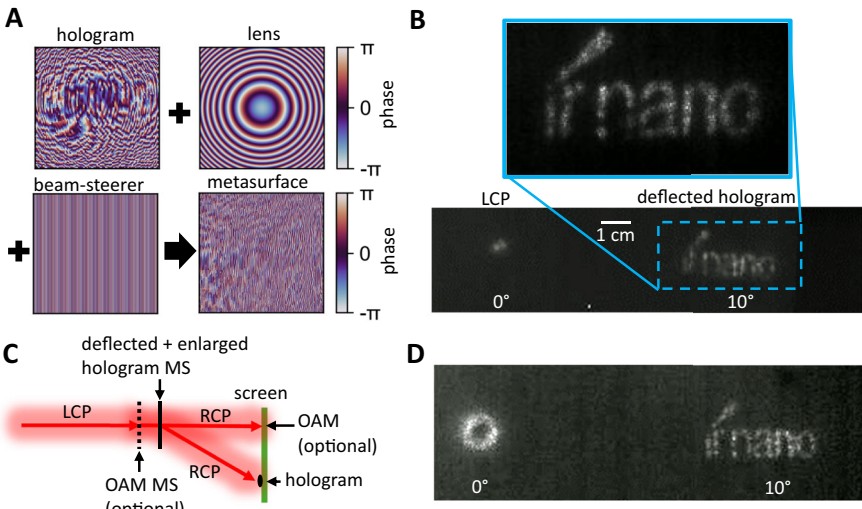

**Fig. 5 | Multifunctionality and cascaded metasurfaces. A** Superimposed phase masks of the hologram, magnifying lens, and beam steerer leading to the phase mask of the metasurface for enlarged and deflected holography. **B** Measured intensity profiles of the corresponding metasurface. The incident LCP light is attenuated and visible at 0°, while the "ir nano" hologram appears deflected at 10°. **C** Sketch of the measurement setup for cascaded metasurfaces creating the OAM at 0° (optional) and the deflected and enlarged hologram at 10°. **D** Measured OAM intensity profile at 0° and deflected hologram at 10°.

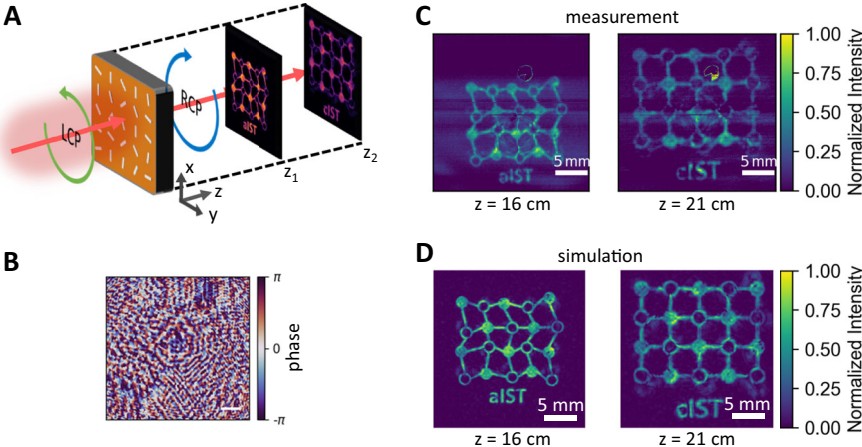

**Fig. 6 | Dual-hologram. A** Schematic sketch of the employed dual-holography concept. The IST metasurface leads to two different holograms at specific distances behind the metasurface. **B** Applied phase mask computed with a diffractive neural network. **C** Measurement of the holograms displaying "aIST" at $z_1 = 16$ cm with a distorted lattice structure and "cIST" with a periodic lattice at $z_2 = 21$ cm. **D** Numerical simulations clearly resolve the two holograms at different distances behind the metasurface.

applications such as beam steering, lensing, and holography. We emphasize IST as a versatile platform for active metasurfaces based on the geometric phase. While the functionality of the designed metasurfaces is entirely different depending on the employed phase mask, the manufacturing process is always the same. In each case, IST rod antennas are optically crystallized via direct laser writing, only the orientation of the antennas is different due to the targeted phase value of each antenna. The concept of directly programming metasurfaces with IST antennas is much simpler, cost-effective, and faster compared to cumbersome fabrication techniques such as conventional lithography involving multiple complex mask designs and costly etching steps (see Supplementary Note 11 for a detailed comparison). The operation wavelength of the metasurfaces is only given by the length of the rotated antennas and can be easily scaled to target the infrared range down to 5 µm or even the terahertz range as long as the operation wavelength exceeds the plasma wavelength of crystalline IST around 900 nm. Commonly, the infrared spectral range displays a significant lack of commercially available beam-shaping elements. Hence, our demonstrated concept paves the way towards rapid prototyping and simple production of reconfigurable meta-optics in the infrared, even for industrial purposes. A comparison of our work with literature about active metasurfaces can be found in Supplementary Note 2.

Customized infrared metasurfaces can be employed in security applications, beam shaping for material processing, or quantum manipulation and information encoding via exploiting the orbital angular momenta of photons[42,43]. Reprogramming the metasurface and modifying the phase profile features a convenient way for replacing spatial light modulators and diffractive optical elements[44].

## Methods
### Sample fabrication
Direct current magnetron sputtering with an LS320 by von Andenne sputter system is used to deposit a 100 nm thin amorphous $In_3SbTe_2$ (IST) film on top of $1 \times 1$ cm² infrared transparent $CaF_2$ substrates. Accordingly, a power of 21 W is applied to an IST target with 99.99 % purity at a base pressure of $6.2 \cdot 10^{-3}$ mbar. Afterward, a 50 nm thin layer of $(ZnS)_{80}:(SiO_2)_{20}$ is deposited with radio frequency magnetron sputtering at a power of 60 W. The deposition time for the IST layer and the capping layer are 19.5 and 32.2 min, respectively. The capping layer is utilized to prevent the samples from oxidization and to facilitate the crystallization process as an anti-reflection coating for the switching laser.

### Optical switching
Local crystallization of the individual crystalline IST antennas is done with the direct laser writing system Photonic Professional GT from Nanoscribe. Here, a 100 fs pulsed laser with a central wavelength of 780 nm and a repetition rate of 80 MHz is employed. The laser pulses are focused by a 63x objective with a numerical aperture of 1.4 onto the sample. The high numerical aperture is achieved by employing the oil 3-(Trimethoxysilyl)propyl methacrylate. Precise movements of the laser are enabled by galvo mirrors with a writing field of $100 \times 100$ µm². Coarse movements of the sample are retrieved by a motorized stage (range of several cm).

For all metasurfaces, we operated the system in the Continuous Mode with a scan speed of 3500 µm/s and a laser power of 15 mW.

The writing time for the beam steering metasurfaces consisting of 1 million antennas was 2.5 h, while the writing time of the metasurfaces with 4 million antennas took about 8 h. A video demonstration of the optical writing process is shown in Supplementary Video 2.

### FTIR measurements
The measured transmittance spectra are recorded by a Bruker Vertex 70 interferometer connected to a Bruker Hyperion 2000 microscope. We applied a 15x Cassegrain objective with a numerical aperture of 0.4 featuring an angular distribution from 10 to 24 degrees. The spectra are recorded with 1000 scans and a resolution of 4 cm⁻¹.

### Metasurface characterization
The fabricated metasurfaces are characterized by a home-built setup. A quantum cascade laser from Daylight Solutions with a wavelength of 9 µm and vertical polarization is circularly polarized with a quarter-wave plate from Optogama designed for a wavelength of 9 µm and then directed onto the metasurface. The initial chirality is filtered out by a second quarter-wave plate in combination with a linear polarizer after passing the metasurface. For detection, different systems were employed. The beam steering metasurfaces (see Fig. 1) are characterized by a mercury cadmium telluride (MCT) detector positioned on a mechanical micrometer stage movable along a semi-circle. For the measurements of the metalens (see Fig. 2), we employed a knife-edge razor blade positioned on a micrometer stage at different positions behind the metasurface and measured the change in the detected laser power. The simple hologram and the orbital angular momentum metasurfaces (see Figs. 3, 4) are projected on a screen and imaged with a FLIR T335 thermal camera. The spiral intensity pattern of the OAM metasurface caused by direct interference with the incident light is

obtained by rotating the linear polarizer to achieve approximately similar laser powers of the converted RCP and incident LCP light. For the dual-hologram (see Fig. 6), we employed the Pyrocam IV by Ophir Photonics to measure the beam intensity profile after passing the metasurface directly. Schematic sketches and more detailed explanations of the different measurement setups can be found in Supplementary Note 12.

## Simulations

Numerical simulations of the antenna spectra and the beam steering metasurface are done with the commercially available program CST Studio Suite from Dassault Systems. The permittivity of IST is taken from Supplementary Note 1. For the $CaF_2$ substrate and the capping layer, a constant refractive index of 2.1 and 1.4 are assumed, respectively. Floquet mode ports are chosen to excite the simulated structures. Unit cell boundaries in lateral dimensions and open boundaries in vertical dimensions are applied. The simulations for the metalens, the hologram and the orbital angular momentum metasurface are done with the freely available Python package LightPipes. Here, the phase profile of the respective metasurface is impinged by a Gaussian beam and then the far-field intensity is calculated for varied distances behind the metasurface. The divergence of the incident laser beam in the experiment is taken into account. The calculation of the dual-hologram metasurface is described in Supplementary Note 10.

## Reporting summary

Further information on research design is available in the Nature Portfolio Reporting Summary linked to this article.

## Data availability

All key data supporting the findings of this study are included in the main article and its Supplementary Information. Additional data sets and raw measurements are available from the corresponding author.

## Code availability

The source code for the calculations conducted in this study is available from the corresponding authors on request.

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

## Acknowledgements
The authors thank Maike Kreutz for the sputter deposition of the thin film layer stack. The authors acknowledge support by the Deutsche Forschungsgemeinschaft (DFG No. 518913417 (L.C. and T.T.) and SFB 917 "Nanoswitches" (L.C., M.W., and T.T.).

## Author contributions
L.C., A.M., R.S., and T.T. conceived the research idea; L.C. and F.B. designed the research; F.B. and A.M. carried out the optical switching; F.B. performed the metasurface measurements. L.C. and F.B. analyzed the data and carried out the numerical simulations. P.B. and C.H. calculated the phase mask of the dual-hologram. M.W. provided the sputtering equipment and phase-change material expertise; all authors contributed to writing the manuscript.

## Funding

## Competing interests
The authors declare no competing interests.
