## [Transparent Peer Review file · Nature Communications]

Infrared Beam-shaping on Demand via Tailored Geometric Phase Metasurfaces employing the Plasmonic Phase-Change Material In_3SbTe_2

Corresponding Author: Mr Lukas Conrads

Version 0:

Reviewer comments:

Reviewer #1

(Remarks to the Author)

In this manuscript, the authors have proposed and experimentally demonstrated direct programming of geometric phase metasurfaces consisting of rotated crystalline In_3SbTe_2 (IST) rod antennas within the plasmonic phase change material (PCM) IST for infrared beam-shaping. Numerous applications, such as beam steering, lensing, holography and orbital angular momenta generating have been achieved. Compared with cumbersome fabrication techniques such as conventional lithography involving multiple complex mask design and costly etching steps, the proposed approach is much simpler, cost-effective, and faster. The demonstrated concept has the potential to facilitate the realization of rapid prototyping and simple production of customized infrared meta-optics for sensing, imaging and quantum information. However, I have several questions as follows:

1. Please provide more information about the fabricated samples, including the period, length, width and height of the unit cell.
2. So many applications, such as beam steering, lensing, holography and so on, are demonstrated in the paper, why? Are there any differences in the manufacturing process or method for these applications?
3. The author mentions that their method has advantages over traditional lithography. However, the statement lacks a comparison to other techniques. In my opinion, the fabrication of the metasurfaces operated in the infrared region is not very difficult and conventional direct laser writing can achieve it.
4. Can the proposed method be used to fabricate metasurface devices for other wavelengths, for example, the visible light?
5. In addition, can this method be applied to other materials? This is because the energy efficiency of the proposed device offers no advantage over commonly used silicon in the infrared region.

Reviewer #2

(Remarks to the Author)

The authors developed a fast-processed metasurface using material properties of In_3SbTe_2 and laser writing system. Through this, they developed a metasurface with steering, lensing, holography, and a combination of these. The results of the developed metasurface seem to be consistent with the authors' arguments, and the simulation results also seem to be quite consistent. However, the novelty of reconfigurable metasurfaces with In_3SbTe_2 is not clear. This manuscript needs major revisions and should be reconsidered.

1. A technological review and comparison with many prior arts are highly needed to clarify novelties of In_3SbTe_2 . There are already many similar works of reconfigurable metasurfaces with phase change materials like VO_2 , GST, GSST. (e.g. A fabrication-free and field-programmable photonic meta-canvas, *Advanced Materials*, 30, 1703878 (2018); Broadband transparent optical phase change materials for high-performance nonvolatile photonics, *Nature Communications*, 10, 4279 (2019); Reconfigurable all-dielectric metalens with diffraction-limited performance, *Nature Communications*, 12, 1225 (2021)). The comparison features may include operation wavelength, index contrast, switching mechanism, switching time, focusing efficiency, and fabrication process.
2. A switching time (a video demonstration would be needed) and operation procedure should be described in detail.
3. The measured optical properties (n & k) should be displayed in the main figure.

4. The In₃SbTe₂ film growth method should be described in detail.
5. Since it is a key strength of the metasurface developed by the high-speed process, it seems necessary to compare how fast it has become to the existing method.
6. What is the definition of ϵ^{\wedge} in Figures s1 and what does the graph mean?
7. How did you get the data for Supplementary Note 1? (calculations, or experiments)
8. What is the width of the antenna with a length of 2.5 μm on page 4?
9. Is there a reason why a 30 μm metasurface with a larger period in Figure 1c has a higher RCP intensity than an 18 μm metasurface?
10. Why metalens focal length is set to 11.5 cm on page 7?
11. Is equation 4 correct in the sentence (The period between adjacent antennas is set to 4 μm , and the exact orientation of each antenna at different positions on the metasurface is calculated with equation 4) on page 7?
12. In the case of meta-lenses, if the resonance wavelength of 9 μm is used, the transmittance of the lens will be lowered, so the efficiency of the lens will not be good, why did you use 9 μm as the operating wavelength, and how much is the lens focusing efficiency (conversion efficiency)?
13. Is there an advantage as a material for meta-lens of In₃SbTe₂ only compared to other PCMs materials?

Version 1:

Reviewer comments:

Reviewer #1

(Remarks to the Author)

The authors have answered all my questions. I recommend acceptance in its current form.

Reviewer #2

(Remarks to the Author)

I appreciate the authors' careful attention to the points raised in the previous review. The revisions were comprehensive and clearly reflect a effort to enhance the clarity of the manuscript. Overall, the clarity and completeness of the manuscript have been significantly improved, and all raised concerns have been adequately addressed. Therefore, I believe the manuscript is suitable for publication in its current form without further revisions.

Infrared Beam-shaping on Demand via Tailored Geometric Phase Metasurfaces employing the Plasmonic Phase-Change Material In_3SbTe_2

Responses to the reviewers' comments and a summary of the changes made to the revised manuscript:

We would like to thank the editor and the reviewers for their review of our work. In this letter, we provide point-by-point responses to each reviewer's comments. The implemented changes are highlighted in the revised marked copy of the manuscript.

Reviewer #1

Reviewer's general statement:

In this manuscript, the authors have proposed and experimentally demonstrated direct programming of geometric phase metasurfaces consisting of rotated crystalline In_3SbTe_2 (IST) rod antennas within the plasmonic phase change material (PCM) IST for infrared beam-shaping. Numerous applications, such as beam steering, lensing, holography and orbital angular momenta generating have been achieved. Compared with cumbersome fabrication techniques such as conventional lithography involving multiple complex mask design and costly etching steps, the proposed approach is much simpler, cost-effective, and faster. The demonstrated concept has the potential to facilitate the realization of rapid prototyping and simple production of customized infrared meta-optics for sensing, imaging and quantum information. However, I have several questions as follows:

Our response:

We thank the reviewer for the detailed summary of our manuscript and highlighting the potential to facilitate rapid prototyping and simple production of customized infrared meta-optics. We will address the questions in the following.

Reviewer's comment 1:

Please provide more information about the fabricated samples, including the period, length, width and height of the unit cell.

Our response:

We thank the reviewer for pointing out this missing information.

Actions taken:

- We included the information in the main text and the methods section:

The period between individual antennas is set to $4\ \mu\text{m}$ with an antenna length and width of $2.5\ \mu\text{m}$ and $0.7\ \mu\text{m}$, respectively. The height of the antennas is limited by the thickness of the IST layer, resulting in a maximum antenna height of $100\ \text{nm}$.

Reviewer's comment 2:

So many applications, such as beam steering, lensing, holography and so on, are demonstrated in the paper, why? Are there any differences in the manufacturing process or method for these applications?

Our response:

We demonstrated all the different applications to highlight IST as a versatile platform for active metasurfaces based on the geometric phase to achieve many different functionalities. While the functionality of the designed metasurfaces is entirely different depending on the employed phase mask, the manufacturing process is always the same. In each case, IST rod antennas are optically

crystallized via direct laser writing, only the orientation of the antennas is different due to the targeted phase value of each antenna. This summarizes the main advantage of the concept: Customized functionalities are directly obtained from varying the orientation of the antenna and imprinting the geometric phase profile, without the need for cumbersome designing and fabrication processes.

Actions taken:

- We highlight IST as a versatile platform for active metasurfaces in the discussion:

We emphasize IST as versatile platform for active metasurfaces based on the geometric phase. While the functionality of the designed metasurfaces is entirely different depending on the employed phase mask, the manufacturing process is always the same. In each case, IST rod antennas are optically crystallized via direct laser writing, only the orientation of the antennas is different due to the targeted phase value of each antenna.

Reviewer's comment 3:

The author mentions that their method has advantages over traditional lithography. However, the statement lacks a comparison to other techniques. In my opinion, the fabrication of the metasurfaces operated in the infrared region is not very difficult and conventional direct laser writing can achieve it.

Our response:

We agree with the reviewer that a proper comparison highlighting the benefits of our method compared to other techniques is missing. While traditional fabrication techniques involve a multitude of different process steps including resist masks deposition and patterning, development and etching, our approach is basically a single-step fabrication technique. Moreover, we would like to emphasize that employing the plasmonic PCM IST offers the ability to perform post-fabrication adaptations of once written nanostructures to modify the functionality and tune the efficiency of the metasurface (see also Conrads et al. Adv. Opt. Mat. 11, 2202696 (2023)). These post-fabrication adaptations are not possible with conventional fabrication techniques and materials, where the functionality is fixed after fabrication.

Actions taken:

- We added a detailed comparison in Supplementary Note 11:

Supplementary Note 11: Comparison with conventional fabrication techniques

Conventionally, the fabrication of optical metasurfaces is a complex and time-consuming process. The schematic principle of a fabrication process via electron beam lithography or laser lithography is displayed in **Figure S12A**. The substrate material is covered with a polymer resist (i) for example via spin-coating. Afterwards, electrons or a laser modifies the resist (ii), which is subsequently removed with a developer (iii) to create the targeted mask of the structures. In a next step, the target material is deposited onto the previously obtained mask (iv). Finally, the remaining resist also covered with the target material is removed in a lift-off process (v). The fabrication of more complex structures requires even multiple repetitions and aligning steps of the previously described procedure.^{15,16}

Even more cumbersome is deep ultraviolet lithography, involving multiple etching steps to finally achieve the targeted nanostructures (see **Figure S12B**).¹⁶

The overall time to fabricate metasurfaces with the previously discussed approaches is estimated as follows: several hours for patterning the sample with the electron beam or laser, 30 minutes to 1 hour for the development, 1-2 hours for the metal deposition and subsequent 30 minutes for the lift-off process. Each additional fabrication step increases the allocated time for fabrication. If now the availabilities of the different machines are taken into account, up to several days are often required until the metasurface is fabricated.

In contrast, our approach of direct optically programming functional metasurfaces is much simpler (c.f. **Figure S12C**). Here, a thin layer of amorphous IST is deposited onto the substrate (ii) and subsequently crystallized via laser irradiation (iii), leading to plasmonic nanostructures directly written into a dielectric surrounding. Moreover, employing IST allows for post-fabrication adaptations of once written nanoantennas by locally addressing the antenna ends with precise laser pulses.^{1,17–19} This is not possible for conventional fabrication techniques where the size and shape of metallic or dielectric nanoantennas are fixed after fabrication.

In summary, our proposed concept speeds up fabrication and prototyping of metasurfaces by omitting time-consuming etching and developing steps. While the required energy to crystallize IST is comparable to other direct laser writing techniques for patterning resist masks, significant energy and costs can be saved by omitting the subsequent processing steps.

Figure S12: Comparison of different fabrication techniques of metasurfaces. A) Conventional metasurface fabrication such as electron beam lithography involves the deposition of a resist mask (i) with subsequent patterning the resist via electrons (e-) or with a laser (ii). Afterwards, the resist developed (iii), leading to a positive or negative mask. After deposition of the target material (iv), the remaining resist with the material is removed with a lift-off or etching process (v). Complex structures involve multiple repetitions of the described procedure. **B)** Deep ultraviolet lithography involves even more steps, including chromium deposition (i), resist deposition (ii), irradiation of the resist (iii), developing the modified resist (iv), dry etching chromium (v), removing the remaining resist (vi), dry etching the substrate (vii) and finally removing the remaining chromium (viii). **C)** Our approach of fabricating functional metasurfaces involves only the deposition of an amorphous IST layer (i) with subsequent direct optical programming of the plasmonic crystalline IST antennas (ii).

Reviewer’s comment 4:

Can the proposed method be used to fabricate metasurface devices for other wavelengths, for example, the visible light?

Our response:

The operation wavelength of the metasurface is only given by the resonance wavelength of the crystalline IST rod antennas which can be easily modified by changing the antenna length. Electric dipole resonances of crystalline IST rod antennas have been already demonstrated from 5 μm to 12 μm (see Heßler et al. Nat. Commun 12, 924 (2021)). We are currently exploring the limitations for smaller antenna structures with sophisticated spatially overlapping laser pulses. However, the plasma frequency of crystalline IST is at around 900 nm; consequently, IST is not metallic in the visible

anymore. IST displays even strong plasmonic behaviour in the terahertz regime around 1 THz (see Zeng et al. *Adv. Opt. Mat.* 11, 2202651 (2023)), allowing for functionalized metasurfaces in that frequency regime. Therefore, the proposed method is limited to the infrared spectral range and larger wavelengths.

Actions taken:

- We clarified to accessible wavelength range in the discussion section:

The operation wavelength of the metasurfaces is only given by the length of the rotated antennas and can be easily scaled to target the infrared range down to 5 μm or even the terahertz range as long as the operation wavelength exceeds the plasma wavelength of crystalline IST around 900 nm.

Reviewer's comment 5:

In addition, can this method be applied to other materials? This is because the energy efficiency of the proposed device offers no advantage over commonly used silicon in the infrared region.

Our response:

The method of direct programming functional metasurfaces in the infrared requires the non-volatile insulator-to-metal transition of the plasmonic PCM IST. Conventional dielectric PCMs such as GST only feature a refractive index contrast or slightly plasmonic behavior in the visible range due to interband transitions (see Gholipour et al. *NPG Asia Mater* 10, 533-539 (2018)). Hence, the reviewer is correct that our method for programmable infrared metasurfaces would also work with other materials featuring a non-volatile insulator-to metal transition as characteristic for IST.

We are not sure how to interpret the mentioned energy efficiency and the comparison to silicon. If the reviewer refers to the energy required to crystallize IST in comparison to the patterning of lithography masks, the reviewer is correct that our method is comparable. Conventional direct laser writing is employed to fabricate polymer masks and subsequently adding the material of interest onto the mask and performing a lift-off or etching process step. However, our method is fundamentally different from these. Here, we exploit the insulator-to-metal transition of IST upon laser heating, which is a single step fabrication (see also new Supplementary Note 11).

If the reviewer refers to the polarization conversion efficiency of the geometric phase metasurfaces, we achieve very similar efficiencies compared to previously reported works employing gold antennas (10% efficiency) (see Yin et al. *Light Sci Appl* 6, e17016 (2017)).

Actions taken:

- We highlight the required energy for local crystallization in Supplementary Note 11:

In summary, our proposed concept speeds up fabrication of metasurfaces by omitting time-consuming etching and developing steps. While the required energy to crystallize IST is comparable to other direct laser writing techniques for patterning resist masks in the range of several 10 mW, significant energy and costs can be saved by omitting the subsequent processing steps.

- We highlight the comparable metasurface efficiency in Supplementary Note 2:

All previously demonstrated concepts in literature require either multiple mask patterning and etching steps⁶⁻⁸, or providing a constant temperature⁶ to omit switching back in the original state³. The efficiency of our metasurfaces is comparable to other works employing plasmonic geometric phase metasurfaces and is limited mainly by the polarization conversion. Cascading different metasurfaces as demonstrated in Figure 5 in the main manuscript increases the efficiency by further converting the remaining incident polarization. The efficiency could be also enhanced by increasing the antenna

density to establish smoother phase gradients, or employ sophisticated antenna designs and layerstacks to suppress reflection and minimize transmitted incident chirality.^{9,10}

The multitude of different beam-shaping applications shown in our manuscript excels all previous work mostly focusing on a single functionality.

Reviewer #2

Reviewer's general statement:

The authors developed a fast-processed metasurface using material properties of In₃SbTe₂ and laser writing system. Through this, they developed a metasurface with steering, lensing, holography, and a combination of these. The results of the developed metasurface seem to be consistent with the authors' arguments, and the simulation results also seem to be quite consistent. However, the novelty of reconfigurable metasurfaces with In₃SbTe₂ is not clear. This manuscript needs major revisions and should be reconsidered.

Our response:

We thank the reviewer for highlighting the consistency of the experiments demonstrated with explanations and simulations. Moreover, we would like to emphasize that, until now, IST has been only exploited for antenna resonance tuning on small scales and tailoring the thermal emission of samples. The novelty of our work is establishing IST as a versatile platform for active nanophotonics by exploiting direct programming of geometric phase metasurfaces to achieve arbitrary control of the scattered light for beam steering, lensing, holography and modifying the orbital angular momentum of photons.

Reviewer's comment 1:

A technological review and comparison with many prior arts are highly needed to clarify novelties of In₃SbTe₂. There are already many similar works of reconfigurable metasurfaces with phase change materials like VO₂, GST, GSST. (e.g. A fabrication-free and field-programmable photonic meta-canvas, *Advanced Materials*, 30, 1703878 (2018); Broadband transparent optical phase change materials for high-performance nonvolatile photonics, *Nature Communications*, 10, 4279 (2019); Reconfigurable all-dielectric metalens with diffraction-limited performance, *Nature Communications*, 12, 1225 (2021)). The comparison features may include operation wavelength, index contrast, switching mechanism, switching time, focusing efficiency, and fabrication process.

Our response:

Our concept of direct laser writing plasmonic metasurfaces facilitates the fabrication process which would otherwise require several cumbersome and time-consuming mask and etching steps. In contrast to conventional PCMs such as GST and GSST featuring only a refractive index contrast upon phase-change, IST switches from an amorphous dielectric phase to a metallic crystalline phase. Therefore, IST can be employed for direct optically writing of functional metasurfaces by locally crystallizing IST. This is in strong contrast to other PCMs, which would only allow for modifying prepatterned metallic antenna resonances by changing the dielectric surrounding of those antennas. In comparison to the volatile phase transition material VO₂, IST remains in its current phase even after the external stimulus, i.e. elevated temperature, is removed. This behavior is characteristic for IST and unambiguously distinguishes IST from VO₂.

Actions taken:

- We compare our work with previously published work in Supplementary Note 2:

Moreover, we compare our demonstrated work with literature about active metasurfaces to clearly highlight the differences. The results are shown in Table S1.

Table S1: Comparison with active metasurfaces in literature.

reference	metasurface	active material	operation wave-length	fabrication process	switching mechanism	switching time	efficiency
this work	beam steering lensing vortex beams holography	In ₃ SbTe ₂	9 μm	direct programming (single step fabrication)	Laser switching	-	13%
Dong et al. ³	beam steering vortex beams holography	VO ₂	10.6 μm	direct programming (constant heat required)	Laser switching	-	1.2%
Zhang et al. ²	Integrated photonic switch	Ge ₂ Sb ₂ Se ₄ Te ₁	1.5 μm	Multiple etching and patterning steps	Electric switching	-	-
Shalaginov et al. ⁴	lensing	Ge ₂ Sb ₂ Se ₄ Te ₁	5.2 μm	Multiple etching and patterning steps	annealing	30 min	24%
Karst et al. ⁵	beam steering	PEDOT:PSS	2.65 μm	Etching and patterning	Electric switching	33 ms	40%
Galaretta et al. ⁶	beam steering	Ge ₂ Sb ₂ Te ₅	1.5 μm	Lithography mask and lift-off	annealing	-	30%
Yin et al. ⁷	beam steering lensing	Ge ₃ Sb ₂ Te ₆	3.1 μm	Lithography mask and lift-off	annealing	2 min	5-10%
Abdollahramezani et al. ⁸	beam steering	Ge ₂ Sb ₂ Te ₅	1.5 μm	Lithography mask and lift-off	Electrical switching	200 ns – 200 μs	80%

All previously demonstrated concepts in literature require either multiple mask patterning and etching steps⁶⁻⁸, or providing a constant temperature to omit switching back in the original state³. While these examples mostly demonstrate switchable functionalities by incorporating active materials, our concept showcases fast and flexible design and fabrication of metasurfaces with different functionalities. The efficiency of our metasurfaces is comparable to other works employing plasmonic geometric phase metasurfaces and is limited mainly by the polarization conversion. Cascading different metasurfaces as demonstrated in Figure 5 in the main manuscript increases the efficiency by further converting the remaining incident polarization. The efficiency could be also enhanced by increasing the antenna density to establish smoother phase gradients, or employ sophisticated antenna designs and layerstacks to suppress reflection and minimize transmitted incident chirality.^{9,10}

The multitude of different beam-shaping applications shown in our manuscript excels all previous work mostly focusing on a single functionality.

Reviewer's comment 2:

A switching time (a video demonstration would be needed) and operation procedure should be described in detail.

Our response:

We thank the reviewer for this excellent suggestion to include a video demonstration highlighting the fast fabrication process of the metasurfaces.

Actions taken:

- We recorded the writing process in Supplementary Video 2 and mentioned the video in the Methods section:

A video demonstration of the optical writing process is shown in Supplementary Video 2.

Reviewer's comment 3:

The measured optical properties (n & k) should be displayed in the main figure.

Our response:

We agree with the reviewer that the optical properties of IST should be included in the main manuscript. The key property of IST is the metallic behavior in the crystalline phase in contrast to dielectric behavior in the amorphous phase. Therefore, we display the complex permittivity instead of the complex refractive index to highlight this behavior more clearly.

Actions taken:

- We included the dielectric permittivity in Figure 1 as new panel B:

The real part of the permittivity for amorphous and crystalline IST is shown in **Figure 1B** (see **Supplementary Note 1** for more details).

Figure 1: [...] **B)** Real part of the permittivity for amorphous and crystalline IST. In the crystalline phase, IST shows metallic behavior ($\epsilon' < 0$).

Reviewer's comment 4:

The In_3SbTe_2 film growth method should be described in detail.

Our response:

We thank the reviewer for his comment and included a more detailed description about the IST sputtering process.

Actions taken:

- We revised the sample fabrication section within the methods:

Direct current magnetron sputtering with an LS320 by *von Andenne* sputter system is used to deposit a 100 nm thin amorphous In_3SbTe_2 (IST) film on top of $1 \times 1 \text{ cm}^2$ infrared transparent CaF_2 substrates. Accordingly, a power of 21 W is applied to an IST target with 99.99 % purity at a base pressure of $6.2 \cdot 10^{-3}$ mbar. Afterwards, a 50 nm thin layer of $(\text{ZnS})_{80}:(\text{SiO}_2)_{20}$ is deposited with radio frequency magnetron sputtering at a power of 60 W. The deposition time for the IST layer and the capping layer are 19.5 minutes and 32.2 minutes, respectively. The capping layer is utilized to prevent the samples from oxidation and to facilitate the crystallization process as an anti-reflection coating for the switching laser.

Reviewer's comment 5:

Since it is a key strength of the metasurface developed by the high-speed process, it seems necessary to compare how fast it has become to the existing method.

Our response:

We agree with the reviewer that a comparison with existing techniques might help the reader to understand the benefits of our concept. Employing the geometric phase of rotated rod antennas allows for fast designing the different functional metasurfaces by only adapting the required phase profile. While the writing time via laser irradiation is comparable to the resist patterning time of other lithography techniques such as electron beam lithography, the most time is usually consumed by the different process steps including development and etching. If also the availabilities of the different machines are taken into account, up to several days are often required until the metasurface is fabricated. Reducing the required fabrication steps to a single step speeds up the fabrication process (see also new Supplementary Note 11). The setup optically writes a 200x200 μm^2 area of the metasurface in approximately 6 seconds, leading to a total fabrication time of 40 minutes for a 4x4 mm^2 large metasurface.

Actions taken:

- We highlight the fast fabrication process omitting multiple steps and the comparison to existing methods in Supplementary Note 11.

Supplementary Note 11: Comparison with conventional fabrication techniques

Conventionally, the fabrication of optical metasurfaces is a complex and time-consuming process. The schematic principle of a fabrication process via electron beam lithography or laser lithography is displayed in **Figure S12A**. The substrate material is covered with a polymer resist (i) for example via spin-coating. Afterwards, electrons or a laser modifies the resist (ii), which is subsequently removed with a developer (iii) to create the targeted mask of the structures. In a next step, the target material is deposited onto the previously obtained mask (iv). Finally, the remaining resist also covered with the target material is removed in a lift-off process (v). The fabrication of more complex structures requires even multiple repetitions and aligning steps of the previously described procedure.^{15,16}

Even more cumbersome is deep ultraviolet lithography, involving multiple etching steps to finally achieve the targeted nanostructures (see **Figure S12B**).¹⁶

The overall time to fabricate metasurfaces with the previously discussed approaches is estimated as follows: several hours for patterning the sample with the electron beam or laser, 30 minutes to 1 hour for the development, 1-2 hours for the metal deposition and subsequent 30 minutes for the lift-off process. Each additional fabrication step increases the allocated time for fabrication. If now the availabilities of the different machines are taken into account, up to several days are often required until the metasurface is fabricated.

In contrast, our approach of direct optically programming functional metasurfaces is much simpler (c.f. **Figure S12C**). Here, a thin layer of amorphous IST is deposited onto the substrate (ii) and subsequently crystallized via laser irradiation (iii), leading to plasmonic nanostructures directly written into a dielectric surrounding. Moreover, employing IST allows for post-fabrication adaptations of once written nanoantennas by locally addressing the antenna ends with precise laser pulses.^{1,17-19} This is not possible for conventional fabrication techniques where the size and shape of metallic or dielectric nanoantennas are fixed after fabrication.

In summary, our proposed concept speeds up fabrication and prototyping of metasurfaces by omitting time-consuming etching and developing steps. While the required energy to crystallize IST is comparable to other direct laser writing techniques for patterning resist masks in the range of several 10 mW, significant energy and costs can be saved by omitting the subsequent processing steps.

Figure S12: Comparison of different fabrication techniques of metasurfaces. A) Conventional metasurface fabrication such as electron beam lithography involves the deposition of a resist mask (i) with subsequent patterning the resist via electrons (e-) or with a laser (ii). Afterwards, the resist developed (iii), leading to a positive or negative mask. After deposition of the target material (iv), the remaining resist with the material is removed with a lift-off or etching process (v). Complex structures involve multiple repetitions of the described procedure. **B)** Deep ultraviolet lithography involves even more steps, including chromium deposition (i), resist deposition (ii), irradiation of the resist (iii), developing the modified resist (iv), dry etching chromium (v), removing the remaining resist (vi), dry etching the substrate (vii) and finally removing the remaining chromium (viii). **C)** Our approach of fabricating functional metasurfaces involves only the deposition of an amorphous IST layer (i) with subsequent direct optical programming of the plasmonic crystalline IST antennas (ii).

Reviewer’s comment 6:

What is the definition of ϵ'' in Figures s1 and what does the graph mean?

Our response:

See response to Reviewer’s comment 7 below.

Reviewer’s comment 7:

How did you get the data for Supplementary Note 1? (calculations, or experiments)

Our response:

We thank the reviewer for pointing out this missing information. We denote the real part of the permittivity to ϵ' and the imaginary part of the permittivity to ϵ'' . In general, the permittivity describes the optical response of materials and is connected to the complex refractive index via $n = \sqrt{\epsilon}$. Since the refractive index does not directly reveals metallic behavior of materials (mainly characterized by a negative real part of the permittivity ($\epsilon' < 0$)), we chose the permittivity as suitable property to characterize the plasmonic PCM IST.

The permittivity data of IST were originally published by Heßler et al. (Nat. Commun 12, 924 (2021)) and are obtained by fitting a Tauc-Lorentz-Drude model to measured infrared reflectance and transmittance data.

Actions taken:

- We added more information about the permittivity data of IST in Supplementary Note 1:

The permittivity of crystalline IST follows a Drude-like behavior with a negative real part of the permittivity ($\epsilon' < 0$). The imaginary part (ϵ'') is zero in the amorphous phase and increases with increasing wavelength for crystalline IST as known from the Drude model. The permittivity values were originally published by Heßler et al.¹ and retrieved by fitting a Tauc-Lorentz-Drude oscillator model to measured infrared spectra. Since the negative real part of the permittivity ($\epsilon' < 0$) directly reveals

plasmonic behavior, we chose the permittivity as suitable property to characterize the plasmonic behavior of IST.

Reviewer's comment 8:

What is the width of the antenna with a length of 2.5 μm on page 4?

Our response:

We thank the reviewer for pointing out the missing details about the fabricated antenna structures.

Actions taken:

- We included the geometric parameters in the main text:

The period between individual antennas is set to 4 μm with an antenna length and width of 2.5 μm and 0.7 μm , respectively.

Reviewer's comment 9:

Is there a reason why a 30 μm metasurface with a larger period in Figure 1c has a higher RCP intensity than an 18 μm metasurface?

Our response:

We attribute the reason for the higher efficiency of the beam steering metasurface with a larger supercell period to the smoother phase gradient for the metasurface with a supercell period of 36 μm . This metasurface consists of 10 differently oriented antennas in contrast to the 5 employed antennas for the beam steering metasurface with the smaller supercell period. Consequently, the phase gradient across adjacent antennas is reduced by a factor of 2 compared to the metasurface with the smaller supercell period. This might lead to an enhancement of the efficiency of the metasurface.

To prove this hypothesis, we performed simulations of a beam steering metasurface with 36 μm supercell period with 10 antennas rotated by 18° to create the 2π phase shift, and another metasurface with 10 antennas where two adjacent antennas (doublets) have the same orientation, and each doublet is rotated by 36°. Therefore, we keep the same amount of antennas and the same supercell period, but tailor the phase gradient across the supercell. The results are shown in Figure R1 and validate the hypothesis that a smoother phase gradient yield a better metasurface efficiency.

Figure R1. Simulated deflected intensity for two beam steering metasurfaces with the same amount of antennas and supercell period, but different phase gradients.

Actions taken:

- We discussed this point in the main text of the manuscript:

We attribute the higher efficiency of the metasurface with a larger supercell period of 36 μm to smaller phase increment steps due to the doubled amount of rotated antennas compared to the metasurface with the supercell period of 18 μm .

Reviewer's comment 10:

Why metalens focal length is set to 11.5 cm on page 7?

Our response:

The choice of the focal length of the metalens to 11.5 cm was arbitrary and mainly motivated to simplify the subsequent measurement by ensuring enough distance between the metasurface and the focal spot. The focal length can be set to every length only dependent on the applied phase profile of the metasurface.

Actions taken:

- We clarified the arbitrary choice of the focal length in the main text:

The large focal distance of the metalens is chosen arbitrarily to ease subsequent measurement by ensuring enough distance between the metasurface and the focal spot.

Reviewer's comment 11:

Is equation 4 correct in the sentence (The period between adjacent antennas is set to 4 μm , and the exact orientation of each antenna at different positions on the metasurface is calculated with equation 4) on page 7?

Our response:

We thank the reviewer for catching this mistake. Of course, the sentence refers to equation 3 to calculate the orientation of the antenna according to the phase profile of the metalens.

Actions taken:

- We included the correct reference to the equation:

The period between adjacent antennas is set to 4 μm , and the exact orientation of each antenna at different positions on the metasurface is calculated with equation 3, leading to a nearly continuously varying phase pattern.

Reviewer's comment 12:

In the case of meta-lenses, if the resonance wavelength of 9 μm is used, the transmittance of the lens will be lowered, so the efficiency of the lens will not be good, why did you use 9 μm as the operating wavelength, and how much is the lens focusing efficiency (conversion efficiency)?

Our response:

We agree with the reviewer that the absorption of the employed nanoantennas at 9 μm wavelength is maximum. However, the scattering of the nanoantennas is maximum as well. Conventionally, the maximum of absorption and scattering coincides at the resonance wavelength (Husnik et al. Physical Review Letters 109, 233902 (2012), Neuman et al. J. Phys. Chem. C 119, 47 (2015)). Therefore, we investigate our fabricated metasurface at 9 μm , leading to maximum scattering of the antennas and maximizing the effect of the metasurfaces. The polarization conversion efficiency for all fabricated metasurfaces is around 13%. Tailoring the design of the antennas, i.e. increasing the anisotropy for enhanced polarization conversion, and increasing the antenna density would lead to higher metasurface efficiencies.

Actions taken:

- We included the reason for the operation wavelength in the main text:

Utilizing the resonance wavelength of the nanoantennas ensures maximum scattering and consequently maximum efficiency of the metasurfaces investigated.

Reviewer's comment 13:

Is there an advantage as a material for meta-lens of In₃SbTe₂ only compared to other PCMs materials?

Our response:

We thank the reviewer for raising this point requiring clarification. Indeed, the proposed concept of direct optical programming geometric phase metasurfaces including the metalens requires a non-volatile insulator-to-metal phase change as present for the plasmonic PCM IST. Other conventionally used PCMs only exhibit a contrast in the refractive index between both phases in the infrared, i.e. the real part of the permittivity is always positive. This is in contrast with the plasmonic PCM IST which can be switched from an amorphous dielectric phase to a crystalline metallic phase (see also Heßler et al. Nat. Commun. 12, 924 (2021), Heßler et al. Nanophotonics 11, 17 (2022), Conrads et al. Adv. Opt. Mat. 11, 2202696 (2023), Conrads et al. Nat. Commun. 15, 3472 (2024)).

Actions taken:

- We included a detailed comparison of the different PCMs in Supplementary Note 2:

Supplementary Note 2: Comparison of IST with other PCMs and usage with active metasurfaces

The plasmonic PCM IST switches from an amorphous dielectric phase to a crystalline metallic one. In the crystalline phase, the permittivity of IST follows a Drude-like behavior (see Supplementary Note 1). Therefore, IST enables direct optical programming of functional metasurfaces by locally crystallizing the PCM with precise laser pulses. This is in strong contrast to conventional dielectric PCMs. Those PCMs are characterized by positive permittivity values in both phases, making dielectric PCMs well suited for metasurface tuning based on a change in the refractive index. **Figure S2** displays the real part (A) and imaginary part (B) of four different materials in the amorphous (solid lines) and crystalline (dashed lines) phases. While conventional dielectric PCMs display an increase of the real part of the permittivity upon crystallization (red curve for Ge₃Sb₂Te₆ (GST) and orange curve for Ge₂Sb₂Se₄Te₁ (GSST)²). The imaginary part for those dielectric PCMs remains nearly zero in both phases. In contrast, the volatile phase-transition material VO₂ and the plasmonic PCM IST feature a negative real part of the permittivity upon crystallization. Therefore, both materials VO₂ and IST show metallic behavior following the Drude-model (also visible in the imaginary part). However, the phase transition of VO₂ is volatile which means that VO₂ remains only metallic if the temperature is above the phase-transition temperature around 68°C. Hence, consistent heating is required to keep the metallic state of VO₂.

Figure S2: Real part (A) and imaginary part (B) of the permittivity for different PCMs. Dielectric PCMs such as GST and GSST (data taken from ref ⁹) feature positive values in the permittivity in both phases. In contrast, IST and VO₂ undergo a metallic transition and follow a Drude-like behavior in the crystalline phase.